# Variable expression of eighteen common housekeeping genes in human non-cancerous kidney biopsies

Philipp Strauss[1]*, Håvard Mikkelsen[1], Jessica Furriol[2]

**1** Department of Clinical Medicine, University of Bergen, Bergen, Norway, **2** Department of Medicine, Haukeland University Hospital, Bergen, Norway

* Philipp.Strauss@uib.no

**Data Availability Statement:** Data is available at GEO (https://www.ncbi.nlm.nih.gov/gds), accession numbers: GSE104948, GSE108113, GSE104954. Additional data is available at https://github.com/pst087-310590-transciptomic-data.

## Abstract

Housekeeping, or reference genes (RGs) are, by definition, loci with stable expression profiles that are widely used as internal controls to normalize mRNA levels. However, due to specific events, such as pathological changes, or technical procedures, their expression might be altered, failing to fulfil critical normalization pre-requisites. To identify RG genes suitable as internal controls in human non-cancerous kidney tissue, we selected 18 RG candidates based on previous data and screen them in 30 expression datasets (>800 patients), including our own, publicly available or provided by independent groups. Datasets included specimens from patients with hypertensive and diabetic nephropathy, Fabry disease, focal segmental glomerulosclerosis, IgA nephropathy, membranous nephropathy, and minimal change disease. We examined both microdissected and whole section-based datasets. Expression variability of 4 candidate genes (*YWHAZ*, *SLC4A1AP*, RPS13 and *ACTB*) was further examined by qPCR in biopsies from patients with hypertensive nephropathy (n = 11) and healthy controls (n = 5). Only *YWHAZ* gene expression remained stable in all datasets whereas SLC4A1AP was stable in all but one Fabry dataset. All other RGs were differentially expressed in at least 2 datasets, and in 4.5 datasets on average. No differences in *YWHAZ*, *SLC4A1AP*, *RPS13* and *ACTB* gene expression between hypertensive and control biopsies were detected by qPCR. Although RGs suitable to all techniques and tissues are unlikely to exist, our data suggest that in non-cancerous kidney biopsies expression of *YWHAZ* and *SLC4AIAP* genes is stable and suitable for normalization purposes.

## 1. Introduction

Housekeeping, or reference genes (RGs) are a group of genes involved in basic cell functions, with a presumed stable expression profile that is independent of cell type and pathophysiological conditions [1]. These RGs are widely used to normalize qPCR data, necessary for robustness and better reproducibility of the results [2–4]. Considering the role played by these technologies in modern research, well-documented normalization strategies are essential.

Since tissue heterogeneity as well as sample quality, isolation and reverse transcription can add variations to final data, normalization is necessary to adjust for the introduced variability.

**Funding:** This project was funded by an open-project grant to Hans-Peter Marti from the Western Norwegian Health Region (Helse vest, project no. 912167). (https://helse-vest.no/en) The funders had no role in study design, data collection and analysis, decision to publish, or preparation of the manuscript.

**Competing interests:** The authors have declared that no competing interests exist.

To be suitable for normalization, RG expression should not display sample variation or correlate with other variables such as treatments, physiological states, gender, age, or sex. Neither should variation occur due to biological changes associated with specific diseases [5].

However, a variety of studies indicate that the expression of several traditional RGs shows considerable variability [5–8]. As a consequence, conclusions drawn from experimental results can point to opposite directions depending on the RG selected for normalization [9]. Therefore, many guidelines suggest prospective testing of selected RGs under the specific conditions required for the planned experiments [2, 9].

While scientifically advantageous, the additional testing is often limited by tissue availability or budget restrictions. To a certain extent this testing can be circumvented, or at least reduced, by studies examining the variability of the RGs in similar tissues. RG testing in cancerous tissues is relatively frequent [10–12], whereas it is less prevalent in non-cancerous renal diseases.

Although common RGs have been validated in diabetic nephropathy [13], various forms of glomerulopathies [7] and allograft tissues [4, 14], the expression of fewer RGs has been verified in hypertensive nephropathy, one of the most common causes of end-stage renal disease (ESRD) in Europe [15]. In recent years, several new RGs have been proposed [3, 11]. While commendable, this effort has deepened the existing problem of insufficient validation, as the newer candidates are often validated to an even lesser extent than the older, often faulty [6, 7, 11], RGs. Considering the uncertainty surrounding RGs, results from non-cancerous kidney diseases urgently require validation.

In recent years the increasing popularity of sequencing technology has resulted in the generation of numerous datasets, that can be mined for data on RGs expression, without performing costly additional experiments [4, 16, 17]. Therefore, here we selected eighteen commonly used RGs, screened them in 30 expression datasets and selected 4 to validate by qPCR in a hypertensive nephropathy and normal kidney biopsies cohort with the aim to identify RGs appropriate for the normalization of RNA data from human non-cancerous kidney samples. We believe that we have achieved that aim.

## 2. Materials and methods

### 2.1 Study design

The study was designed in accordance with MIQE guidelines [2]. RGs were selected based on the frequency of their use across all tissues and in previous investigations, with special emphasis on papers examining gene expression in non-cancerous renal tissue. A flowchart of the study design is depicted in **Fig 1**. Data has been made available in the GitHub data repository (https://github.com) in the repository 310590-transciptomic-data.

Reference genes (RGs) were selected from the literature based on frequency of use, and whether they had previously been evaluated in kidney biopsies from non-cancerous renal tissue. A selection of RGs the expression of which has only been investigated in cancer tissues has also been included. The additional datasets referenced in the last box refer to dataset 9–12.

The following RGs were selected for examination in this investigation: Glyceraldehyde 3-phosphate dehydrogenase (*GAPDH*, ENSG00000111640), Actin gamma 1 (*ACTG1*, ENSG00000184009), REL Proto-Oncogene, NF-KB Subunit (*REL*, ENSG00000162924), Actin beta (*ACTB*, ENSG00000075624), Solute carrier family 4 member 1 adaptor protein (*SLC4AIAP*, ENSG00000163798), Tyrosine 3-Monooxygenase/Tryptophan 5-Monooxygenase Activation Protein Zeta (*YWHAZ*, ENSG00000164924), Ribosomal Protein S13 (*RPS13*, ENSG00000110700), NOP10 Ribonucleoprotein (*NOP10*, ENSG00000182117), Phosphoglycerate Mutase 1 (*PGAM1*, ENSG00000171314), Peptidylprolyl Isomerase A (*PPIA*, ENSG00000196262), Glucuronidase Beta (*GUSB*, ENSG00000169919), TATA-Box Binding

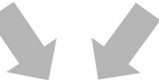

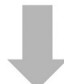

**Fig 1. Workflow.**

Protein (*TBP*, ENSG00000112592), Ribosomal Protein L13 (*RPL13*, ENSG00000167526), Heterogeneous Nuclear Ribonucleoprotein L (*HNRNPL*, ENSG00000104824), Poly (RC) Binding Protein 1 (*PCBP1*, ENSG00000169564), Retention In Endoplasmic Reticulum Sorting Receptor 1 (*RER1*, ENSG00000157916), Phospholipase A2 Group IVA (*PLA2G4A*, ENSG00000116711) and Beta-2-Microglobulin (*B2M*, ENSG00000166710).

Following selection of prospective RGs from the literature, their expression was evaluated in our own and publicly available datasets (see below).

Since a suitable RG should under no circumstances be differentially expressed in control and test samples as it is used as internal control in those groups, we utilized differential expression as a measure of stability. Four candidate genes, including those providing the best results in the 30 datasets comparison and some of the most used RGs were selected and further evaluated by qPCR.

The Regional Ethics Committee (REC) of Western Norway approved the study (REK vest 2013/553). Written informed consent was obtained from all patients whose biopsies were part of our own experiments.

## 2.2 Datasets

A total of 30 datasets were selected. They were acquired from our unpublished data (n = 14), publicly available datasets provided by the European Renal cDNA Bank (ERCB) [18–20] (n = 2) or by the Neptune Network [21] (n = 2). Additionally, we used publicly available datasets (n = 12). A detailed overview, including references and links, of each dataset is provided in **Table 1**. As controls, datasets included biopsies from healthy donors (10 databases), stable allografts (3 databases) or biopsies with minimal and unspecific alterations (12 databases). In our own data normal controls were selected from a group of biopsies graded by the renal-pathologist on duty as "not containing any or only insignificant pathology". We re-examined the biopsies histology and accessed the patients' clinical record. Patients that later developed renal disease, kidney failure or severe autoimmune disease or showed severe proteinuria were excluded. Our own dataset's biopsies were always taken for diagnostic purposes and therefore aimed at the kidney cortex. Biopsies with less than 50% cortex were discarded. Approximately

**Table 1. Dataset details.**

| Data-Set ID | Disease | Source | GEO accession number | Seq. method | Micro-dissected | Compartment | N | Control type |
|---|---|---|---|---|---|---|---|---|
| 1 | MCD | Internal | N.A. | NGS | Yes | Glomeruli | 22 | Healthy control |
| 2 | MN | Internal | N.A. | NGS | Yes | Glomeruli | 20 | Healthy control |
| 3 | HT | Internal | N.A. | NGS | No | N.A. | 12 | Healthy control |
| 4 | DIA2 | Internal | N.A. | NGS | No | N.A. | 12 | Healthy control |
| 5 | Fabry | Internal | N.A. | NGS | Yes | Glomeruli | 16 | Healthy control |
| 6 | Fabry | Internal | N.A. | NGS | Yes | Arteries | 16 | Healthy control |
| 7 | Fabry | Internal | N.A. | NGS | Yes | Proximal tubule | 16 | Healthy control |
| 8 | Fabry | Internal | N.A. | NGS | Yes | Distal Tubule | 16 | Healthy control |
| 9 | MN | ERCB | N.A. | MA | Yes | Glomeruli | 69 | Healthy control |
| 10 | MCD | ERCB | N.A. | MA | Yes | Glomeruli | 62 | Healthy control |
| 11 | MN | Neptune | N.A. | MA | Yes | Glomeruli | 55 | Healthy control |
| 12 | MCD | Neptune | N.A. | MA | Yes | Glomeruli | 54 | Healthy control |
| 13 | Fabry | Internal | N.A. | NGS | Yes | Glomeruli | 16 | Healthy control |
| 14 | Fabry | Internal | N.A. | NGS | Yes | Arteries | 16 | Healthy control |
| 15 | Fabry | Internal | N.A. | NGS | Yes | Proximal tubule | 16 | Healthy control |
| 16 | Fabry | Internal | N.A. | NGS | Yes | Distal Tubule | 16 | Healthy control |
| 17 | MN | Internal | N.A. | NGS | Yes | Glomeruli | 26 | MCD |
| 18 | MN_PLA2R_neg | Internal | N.A. | NGS | Yes | Glomeruli | 12 | MN_PLA2R_pos |
| 19 | RPGN | GEO | GSE104954 | MA | Yes | Tubulointerstitial | 39 | Healthy control |
| 20 | MCD | GEO | GSE104954 | MA | Yes | Tubulointerstitial | 26 | Healthy control |
| 21 | FSGS | GEO | GSE104954 | MA | Yes | Tubulointerstitial | 25 | Healthy control |
| 22 | DIA | GEO | GSE104954 | MA | Yes | Tubulointerstitial | 25 | Healthy control |
| 23 | DIA | GEO | GSE104954 | MA | Yes | Tubulointerstitial | 30 | HT |
| 24 | HT | GEO | GSE104954 | MA | Yes | Tubulointerstitial | 52 | Lupus |
| 25 | DIA | GEO | GSE104954 | MA | Yes | Tubulointerstitial | 35 | IGAN |
| 26 | HT | GEO | GSE104948 | MA | Yes | Glomeruli | 42 | Healthy control |
| 27 | IgA | GEO | GSE104948 | MA | Yes | Glomeruli | 42 | Healthy control |
| 28 | TCMR | GEO | GSE120495 | NGS | No | N.A. | 10 | STA |
| 29 | ATI | GEO | GSE120495 | NGS | No | N.A. | 10 | STA |
| 30 | IFTA | GEO | GSE120495 | NGS | No | N.A. | 10 | STA |

MCD: Minimal change disease, MN: Membranous nephropathy, HT: Hypertension, DN: Diabetes type 2, FSGS; Focal segmental glomerulosclerosis, IGAN; IgA nephropathy, TCMR: t-cell mediated rejection, RPGN; Rapidly progressive glomerulonephritis, STA: stable allograft, ATI: acute tubular injury, IFTA: Interstitial fibrosis and tubular atrophy. GEO; Gene Expression Omnibus, NGS: Next generation sequencing, MA: Microarray

70% of the biopsies had 10 or more glomeruli. All microdissection was performed on the same Zeiss PALM Lasor Capture Microdissection (LCM) system (Carl Zeiss AG, Oberkochen, Germany) with consistent personal and settings for each dataset. After microdissection the samples were immediately stored at -80 degrees till rna extraction, after which they were again immediately stored at -80 degrees.

A total of 5 datasets (n = 54 samples) included whole kidney tissues. Moreover, since microdissection allows refining of input tissue and might reveal differences buried under noise in whole-sections, 25 datasets (n = 764 samples) included microdissected tissues from glomeruli, arteries, proximal or distal tubules, and tubointerstitial structures. In all datasets comparisons were only made within the dataset, we did not compare groups from one dataset to groups from another dataset, and in microdissected datasets we only compared the same

compartments from different patient groups, e.g., hypertensive glomeruli compared to glomeruli from healthy controls, all from the same dataset.

A total of 13 datasets were sequenced via microarray and 17 via next generation sequencing. In particular, 5 datasets included samples from patients with minimal change disease (MCD); 8 from patients with Fabry disease; 5 from patients with membranous nephropathy (MN); 3 from patients with hypertensive nephropathy (HN), and 4 from patients with diabetic nephropathy (DN). Full details on each patient cohort from external data is available through the original publication for each external dataset, see **Table 1.** In internal datasets, patients suffering, at the time of the initial biopsy from concurrent renal failure, cancers or other renal diseases, apart from the primary diagnosis were excluded. All patients were Caucasian. Apart from the Fabry derived datasets all patients were over 18 years old. Across datasets genders approximately equally distributed, with more males present in the Fabry data.

## 2.3 Patient selection for qPCR

Kidney biopsies used for qPCR analysis (n = 16) were selected from the Norwegian Renal Biopsy Registry. Biopsies from patients with hypertensive nephropathy (HT) (n = 11) were compared to normal biopsies or samples with minimal and unspecific changes (n = 5). HT patients were matched to the non-diseased controls (NDC) for age (-/+ 5 years), and sex. Each sample was diagnosed and scored by an experienced renal pathologist. Furthermore, all cases were reassessed prior to inclusion in the study.

Average age was 54 ± 5.5 years old for NDC and 56 ± 4.6 years old for HT patients. HT patients with renal tissue alterations attributable to a different disease were not included.

All biopsies were stored as formalin-fixed and paraffin-embedded (FFPE) tissues at room temperature.

## 2.4 RNA isolation and cDNA synthesis

Two to eight 10 μm thick sections were cut from FFPE blocks and used as input. The number of sections was determined by the surface area covered by tissue in each biopsy. RNA was then isolated as previously described [22], using miRNeasy FFPE kit (cat no. 217504; Qiagen, Venlo, The Netherlands) according to manufacturer's instructions.

Following extraction, samples were stored at -80˚C. RNA concentration was measured with a Qubit RNA BR Assay kit (cat no. Q10210; ThermoFisher) in a Qubit 4 Fluorometer (Q33238; ThermoFisher). The median concentration was 59,8 ng total rna (range 22,6–242). A260/A280 and 260/230 ratios were measured using a NanoDrop One Spectrophotometer (ThermoFisher), with a median of 1,905 (range 1,67–1,98) and 1,85 (range 1,01–2,11) respectively. cDNA synthesis was performed from 200 ng of RNA using SuperScript IV VILO master mix with ezDNase (No. 11766050; Thermo Fisher Scientific).

## 2.5 Quantitative real-time polymerase chain reaction

Quantitative real-time polymerase chain reaction (qPCR) was performed using TaqMan Fast Advanced master mix (No. 4444556; Thermo Fisher Scientific). Technical triplicates were fulfilled for each sample and probe.

The following probes purchased from Thermo Fisher Scientific were used; *RPS13* (Catalog number: 4331182, Hs01011487_g1), *YWHAZ*(Catalog number: 4331182, Hs01122445_g1), *SLC4AIAP* (Catalog number: 4331182, Hs00250835_m1), *ACTB* (Catalog number: 4331182, Hs03023943_g1).

Experiments were performed according to manufacturer's instructions. qPCR was performed on a 7500 fast real-time PCR system (Applied Biosystems, Carlsbad, CA, USA). The

instrument was set to Uracil-N glycosylase incubation at 50˚C for 2 minutes followed by Polymerase activation at 95˚C for 2 minutes. PCR was then performed for 40 cycles with denaturation at 95˚C for 1 second and annealing/extension at 60˚C for 20 seconds. Amplification of each RG was tested in three technical replicates for each sample and negative controls without templates were included in every experiment.

## 2.6 Statistical analysis

Fold changes for the 30 unpublished and publicly available datasets were calculated for the complete data in the R environment, version 1.3.1056, and p-values adjusted with the Benjamini-Hochberg method.

The number of datasets where an RG was differentially expressed in control and test samples were tallied and RGs with the lowest number picked as top candidates. The lowest number was zero, i.e. The RG was not differentially expressed in any dataset. Plots were generated using SPSS (v.25; IBM Corp., Armonk, NY, USA). Correlations were determined using Pearson test and continuous variables for age, and categorical variables for gender and sample group. Significance and p-values from the qPCRs were obtained using the Mann–Whitney U test according to ΔCt values from each sample. Cutoff for significance was set at $p < 0.05$.

## 2.7 Library preparation and Bioinformatics for all datasets

Datasets acquired from ERCB (9 and 10) or the Neptune cohort (11 and 12) were processed as previously described [18–21].

Datasets from our own group concerning patients suffering from Fabry's disease (n = 8; datasets 5–8 and 13–16) were obtained as follows: RNA sequencing libraries were prepared using standard Illumina Access protocol (RNA exome, Illumina, San Diego, CA, USA) on an Illumina platform in different batches due to the large number of samples, at the following genomic facilities: i) the Norwegian University of Science and Technology (NTNU) in Trondheim, Norway, in collaboration with PhD Vidar Beisvåg and his group, ii) Firalis SA, Huningue, France, in collaboration with Eric Schordan, and iii) the Functional Genomics Center Zurich (CHRO), University of Zurich, Switzerland. However, library normalization was performed exclusively at the Norwegian University of Science and Technology, and libraries were normalized to 2.2 pM for the NextSeq500 instrument and 2.3 pM for the HiSeq 4000 instrument.

Samples were subjected to paired-end 2x75 bp sequencing with around 60M paired end reads. Base calling was done on the HiSeq instrument by RTA 1.17.21.3. FASTQ files were generated using bcl2fastq v2.20 (Illumina, Inc. San Diego, CA, USA). Transcript expression values were generated by quasi alignment using Salmon (http://salmon.readthedocs.io/en/latest/index.html) and Ensembl (GRCh38) human transcriptomes. Aggregation of transcript to gene expression was performed using tximport (http://bioconductor.org/packages/release/bioc/html/tximport.html). An empirical expression filter was applied, which left genes with more than 1 counts per million (cpm) in more than 25% of samples per dataset. Comparative analysis was done using voom/Limma R-package.

Differential gene expression in control and test samples was defined as Benjamini-Hochberg adjusted p-value ≤0.05, and an absolute fold change of ≥2. Based on unsupervised clustering and PCA correlation analysis, potential batch effects within the RNAseq data were mitigated using ComBat in combination with CPM-normalization [23]. Subsequently, using a standard DESeq2 workflow, differential gene expression was assessed to compare all groups from the same compartment [24].

Our own datasets concerning Minimal change disease (n = 1, no. 1) and Membranous nephropathy (n = 3, no. 2 and no. 17–18) were processed as follows: RNA library preparation was performed using the TruSeq RNA Access Library Preparation Kit (Illumina, Inc., San Diego, CA, USA). NextSeq500 system (Illumina, Inc., San Diego, CA, USA) was used for RNA sequencing at the Genomics Core Facility, Norwegian University of Science and Technology (NTNU). Assembly of reads was aligned to the Homo sapiens hg38 reference genome using Gencode (https://www.gencodegenes.org/) [25]. Differentially expressed genes (DEGs) with a count per million (CPM) of more than 3 in at least four samples and an absolute fold-change value of greater than 2 and adjusted p-value <0.05 were included in the analysis. Statistical analysis was performed with Limma/Voom package [26].

Sequencing libraries for the diabetic and hypertensive nephropathy datasets from our own group (datasets 3–4) were generated using the TruSeq RNA exome library kit (Illumina, San Diego, CA, USA) according to manufacturers' instructions. Libraries were quantitated by qPCR using the KAPA library quantification kit–Illumina/ABI Prism (Kapa Biosystems, Wilmington, MA, USA) and validated using the Agilent high-sensitivity DNA kit on a bioanalyser. They were subsequently normalized to 2.6 pM and subjected to cluster and paired-end read sequencing, performed for 2× 75 cycles on two NextSeq500 HO flow cells (Illumina), according to manufacturer's instructions. Base-calling was performed using the NextSeq500 instrument, and RTA 2.4.6. FASTQ files were generated using bcl2fastq2 conversion software (v.2.17; Illumina). Assembled reads were aligned to the Homo sapiens hg38 reference genome using Gencode (gencodes.org). Differentially expressed genes (DEGs) with >3 counts per million (CPM) in at least four samples, absolute fold-change (FC) value >2, and adjusted p-value <0.05 were included in the analysis.

Datasets 19–30 were obtained through the Gene Expression Omnibus (GEO). In particular, datasets 19–25 corresponding to GSE104954 [27] were analyzed using the GEO2R analysis tool [28, 29] provided by GEO. Datasets 26–27, corresponding to GSE104948, were used as normalized data. Similarly, for datasets 28–30, corresponding to GSE120495, we used normalized data provided by original authors [4]. Additional details are provided in **Table 1**.

## 3. Results

### 3.1 Reference gene expression variability

Comparison of the 30 different kidney-related gene expression datasets, showed that among commonly used RGs, *SLC4AIAP* and *YWHAZ* were more consistently expressed in control and test samples (**Fig 2**). In particular, *YWHAZ* gene was not differentially expressed in any dataset, whereas *SLC4AIAP* was differentially expressed in controls and test specimens in one dataset (no. 14) including microdissected arteries from patients with longstanding Fabry disease.

Excluding the two top contenders, the number of available datasets showing evidence of variable RG expression in control and test samples ranged between 2/26 (12%) for *PPIA* and 8/26 (31%) for *HNRNPL* (**Figs 2 and 3A**).

On the other hand, notably, *YWHAZ* and *SLC4AIAP* gene expression was undetectable in 3/30 (10%) and 5/21 (24.8%) available databases, respectively. Databases from non-microdissected libraries including stable allograft tissues, as controls, appeared to be peculiarly concerned, as neither *YWHAZ* nor *SLC4AIAP* were detected in any of the three datasets that fulfilled these criteria (Dataset 28–30, see **S1 Table**). However, dataset 28–30 originated from the same experiment and are not independent from each other.

| Dataset ID | Disease | Control type | Micro-dissected | YWHAZ | SLC4A1AP | ACTG1 | GAPDH | REL | PPIA | ACTB | TBP | RPL13 | PLA2G4A | B2M | NOP10 | RPS13 | GUSB | PGAM1 | PCBP1 | RER1 | HNRNPL |
|---|---|---|---|---|---|---|---|---|---|---|---|---|---|---|---|---|---|---|---|---|---|
| 1 | MCD | Norm | Yes | No | No | No | No | No | No | No | No | No | Yes | Yes | No | Yes | Yes | No | Yes | No | Yes |
| 2 | MN | Norm | Yes | No | No | No | No | No | No | No | No | Yes | Yes | Yes | No | Yes | Yes | No | Yes | No | Yes |
| 3 | HT | Norm | No | No | No | No | No | No | No | No | No | No | No | No | No | No | No | No | No | No | No |
| 4 | DIA | Norm | No | No | No | Yes | No | No | No | No | No | No | No | No | No | No | No | No | No | No | No |
| 5 | Fabry | Norm | Yes | No | No | Yes | Yes | No | Yes | Yes | No | Yes | No | Yes | No | Yes | Yes | Yes | Yes | Yes | No |
| 6 | Fabry | Norm | Yes | No | No | No | No | No | No | No | Yes | No | No | No | No | No | No | Yes | No | Yes | No |
| 7 | Fabry | Norm | Yes | No | No | No | No | No | No | Yes | No | No | No | No | No | No | Yes | No | No | No |
| 8 | Fabry | Norm | Yes | No | No | Yes | Yes | No | No | Yes | No | No | No | No | No | No | No | No | Yes | No |
| 9 | MN | Norm | Yes | No | ND | NA | NA | NA | NA | NA | NA | NA | NA | NA | No | ND | NA | NA | NA | NA | NA |
| 10 | MCD | Norm | Yes | No | ND | NA | NA | NA | NA | NA | NA | NA | NA | NA | No | ND | NA | NA | NA | NA | NA |
| 11 | MN | Norm | Yes | No | No | NA | NA | NA | NA | NA | NA | NA | NA | NA | No | Yes | NA | NA | NA | NA | NA |
| 12 | MCD | Norm | Yes | No | No | NA | NA | NA | NA | NA | NA | NA | NA | NA | No | Yes | NA | NA | NA | NA | NA |
| 13 | Fabry | Norm | Yes | No | No | Yes | Yes | No | Yes | Yes | No | Yes | No | No | No | Yes | No | Yes | Yes | No |
| 14 | Fabry | Norm | Yes | No | Yes | No | No | No | No | No | No | No | No | No | No | Yes | Yes | No | No | No |
| 15 | Fabry | Norm | Yes | No | No | No | No | No | No | No | No | No | No | No | No | No | No | No | No | No |
| 16 | Fabry | Norm | Yes | No | No | No | No | No | No | No | No | No | No | No | No | No | No | No | No | No |
| 17 | MN | MCD | Yes | No | No | No | No | No | No | No | No | No | No | Yes | Yes | No | No | No | No | Yes | No |
| 18 | MN | MN | Yes | No | No | No | No | Yes | No | No | No | Yes | No | No | Yes | No | Yes | No | No | No |
| 19 | RPGN | Norm | Yes | No | NA | Yes | No | No | NA | No | Yes | No | Yes | Yes | Yes | NA | No | NA | Yes | No | Yes |
| 20 | MCD | Norm | Yes | No | NA | No | No | No | NA | No | No | No | No | No | No | NA | No | NA | No | No | Yes |
| 21 | FSGS | Norm | Yes | No | NA | No | No | No | NA | No | No | No | No | No | No | NA | No | NA | No | No | Yes |
| 22 | DIA | Norm | Yes | No | NA | No | No | No | NA | No | Yes | No | Yes | No | No | NA | No | NA | No | No | Yes |
| 23 | DIA | HT | Yes | No | NA | No | No | No | NA | No | No | No | No | No | No | NA | No | NA | No | No | No |
| 24 | HT | Lupus | Yes | No | NA | No | No | No | NA | No | No | No | No | No | No | NA | No | NA | No | No | No |
| 25 | DIA | IgA | Yes | No | NA | No | No | No | NA | No | No | No | No | No | No | NA | No | NA | No | No | Yes |
| 26 | HT | Norm | Yes | No | NA | No | No | No | NA | No | No | No | No | Yes | No | NA | No | NA | No | No | No |
| 27 | IgA | Norm | Yes | No | NA | No | No | No | NA | No | No | Yes | No | Yes | No | Yes | NA | Yes | Yes | No |
| 28 | TCMR | STA | No | ND | ND | No | No | No | No | No | No | No | No | No | No | No | No | No | No | No | No |
| 29 | ATI | STA | No | ND | ND | No | No | No | No | No | No | No | No | No | No | No | No | No | No | No | No |
| 30 | IFTA | STA | No | ND | ND | No | No | No | No | No | No | No | No | No | No | No | No | No | No | No | No |

**Fig 2. Results for all included reference genes from each dataset.** MCD: Minimal change disease, MN: Membranous nephropathy, HT: Hypertension, DIA2: Diabetes type 2, FSGS; Focal segmental glomerulosclerosis, IGAN; IgA nephropathy, TCMR: t-cell mediated rejection, RPGN; Rapidly progressive glomerulonephritis, STA: stable allograft, ATI: acute tubular injury, IFTA: Interstitial fibrosis and tubular atrophy. In the columns under the gene IDs "Yes" refers to genes differentially expressed in control and test samples in the dataset. "No" refers to RG equally expressed in control and test samples in the defined dataset. Not available (NA) refers to RG not tested in specific datasets. Not detected (ND) refers to genes undetected in the specific dataset.

## 3.2 Reference gene expression variability in specific datasets

Expression of the RG under investigation was analyzed in each dataset. In 10/30 datasets expression of different tested RG did not show any variation between control and test samples.

However, in the remaining 20 datasets, the expression of 6–56% of the available RG under investigation varied (**Fig 3B**).

Importantly, variation rates did not appear to be obviously associated with defined types of sample preparation, disease or controls. For instance, in databases addressing gene expression in microdissected samples from patients with Fabry disease (n = 8), variations in RG expression ranged between 0 (n = 2) and 56% (n = 1) (**Figs 2 and 3B**). Similarly, RG expression variations in membranous nephropathy databases ranged between 0 (n = 1) and 25% (n = 1). The commonly used RG *GAPDH* was differentially expressed only in 3/8 Fabry disease datasets. Full results including foldchanges and pvalues for each dataset for each RG are provided in **S1 Table**.

## 3.3 qPCR

To validate results from available databases, we examined the expression of *YWHAZ* and SLC4A1AP, the best candidate RGs, in FFPE-derived specimens from patients with hypertension (HT, n = 11) and non-diseased controls (NDC, n = 5). As control RG, we used *ACTB* and *RPS13* genes (**Table 2**). Median A260/A280 ratio of the RNA samples was 1.88 (range 1.67–1,98) consistent with a good quality of the RNA output.

Expression levels of the four RG in combined test and control samples were comparable (**Fig 4A**). More importantly, the expression of each RG did not significantly differ between HT and control specimens (**Fig 4B**). Corresponding p-values are reported in **Table 3A**. Moreover, the expression of the four candidate RGs appeared to be highly correlated ($\geq 0,899$; $p < 10^{-6}$) (**Table 3B**).

## 4. Discussion

In this study we investigated the expression of 18 commonly used RGs in 30 datasets including samples from patients with a wide range of renal diseases other than cancer, aiming at the identification of genes allowing appropriate RNA data normalization.

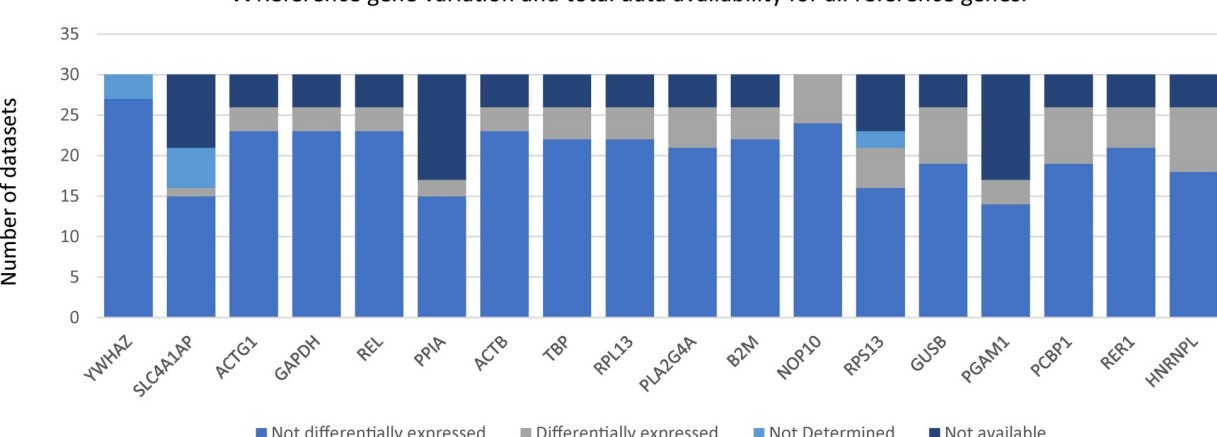

**A** Reference gene variation and total data availability for all reference genes.

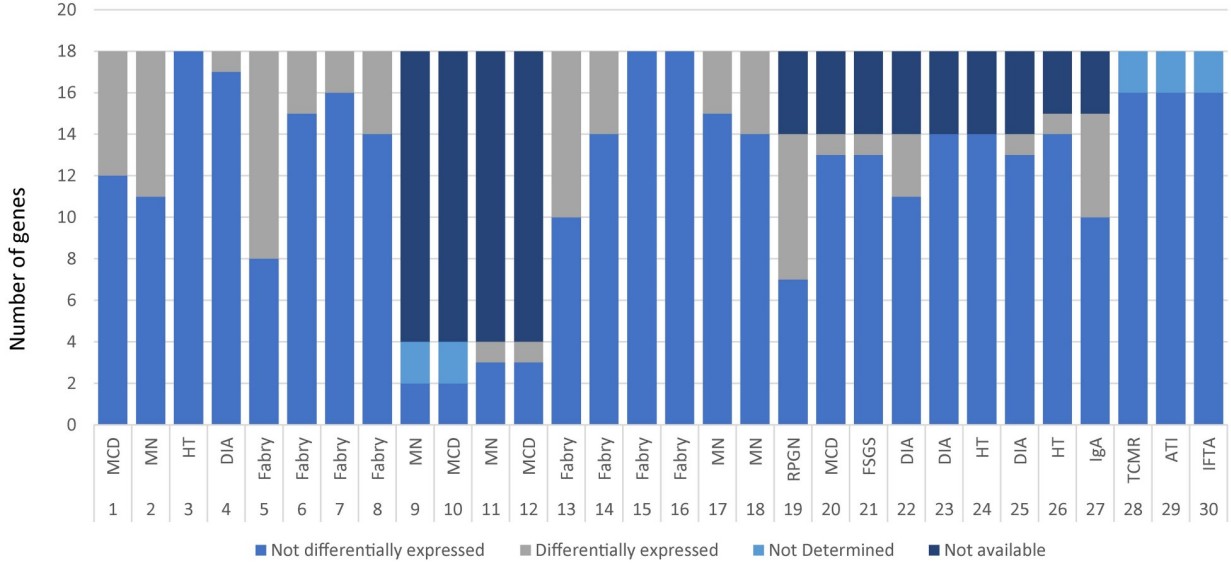

**B** Degree of variation across all reference genes for each database

**Fig 3. Variations in reference gene (RG) expression per gene and per database.** Panel **3A** displays RGs along the x-axis and datasets along the y-axis. For each RG the number of datasets where the RG was either not detected, not available, differentially expressed or not differentially expressed is marked. Not available (NA) refers to RGs not tested in specific datasets. Not detected (ND) refers to genes undetected in the specific dataset. If a particular variable is not listed its value was zero, such as e.g., the number of datasets were YWHAZ was differentially expressed is not listed, since YWHAZ was stable in all datasets. Panel **3B** displays Datasets along the x-axis and RGs on the y-axis. For each database how many RGs were either not detected, not available, differentially expressed or not differentially expressed is marked. Characteristics of each database are described in detail on **Table 1**.

Our main finding is that using any single RG in the analysis of different databases implies the risk of introducing large experimental bias.

We found that *YWHAZ* represents a top RG, with no differences in expression between samples in all datasets where the expression data were available.

The importance of stable RGs can be demonstrated by comparing the results from using stable vs unstable RGs in the same experiment. In a theoretical example, if we were interested

**Table 2. Candidate RG for PCR validation.**

| Gene name | Ensembl | Full name | Biological process | Probes |
|---|---|---|---|---|
| SLC4A1AP | ENSG00000163798 | Solute Carrier Family 4 Member 1 Adaptor Protein | RNA splicing | Hs00250835_m1 |
| ACTB | ENSG00000075624 | Actin Beta | Actin filament fragmentation | Hs03023943_g1 |
| RPS13 | ENSG00000110700 | Ribosomal Protein S13 | Translation | Hs01011487_g1 |
| YWHAZ | ENSG00000164924 | Tyrosine 3-Monooxygenase/Tryptophan 5-Monooxygenase Activation Protein Zeta | Signal transduction | Hs01122445_g1 |

in the expression of PON1 in Fabry's disease, we could perform qPCR to assess the difference between patients with Fabry's disease and healthy controls. In our data PON1 was not affected in Fabry's disease (Fabry vs Normal FC: 0.97). However, if we were to choose GAPDH (Fabry vs Normal FC 0.49) as RG we would have to conclude that PON1 is overexpressed in Fabry's disease, as the GAPDH gene itself is significantly decreased in patients with Fabry disease. Therefore, the normalization will leave PON1 expression artificially higher in the Fabry group, while being decreased normally in the normal controls. If, on the other hand, we use YWHAZ as the RG, the results change. YWHAZ (Fabry vs Normal FC: 1.05) is stable in Fabry's disease, no bias is introduced, and the results show that PON1 is not differentially expressed.

*YWHAZ* encodes a highly conserved protein mediating signal transduction by binding to phosphoserine-containing proteins. It was recently proposed as a "central hub protein for many signal transduction pathways" in a variety of cancers [30], and has been described as unfavorable prognostic marker in renal cancer (https://www.proteinatlas.org/ENSG00000164924-YWHAZ/pathology) [31]. These data suggest that, while *YWHAZ* might be suitable as a RG in non-cancerous renal tissues, caution is warranted on applying it to renal cancer tissues, as previously proposed [32]. In non-cancerous renal tissue, suppression of *YWHAZ* gene expression has resulted in glomerular mesangial cell proliferation in early diabetic nephropathy in primary mouse mesangial cells [33].

SLC4A1AP, encoding a solute carrier protein, might represent an additional interesting RG candidate. However, the expression of this gene was undetectable in 5/21 available databases, thus questioning its potential relevance.

As noted previously, we are not the first to investigate RG variation in non-cancerous renal biopsies. Kidney specific investigations were performed by Schmid et al. [7] who examined the stability of *GAPDH*, 18S rRNA and *PPIA* in 165 renal biopsies from a variety of diseases. Their results for *GAPDH* were unfavorable, while they recommended the use of 18S rRNA and *PPIA*. Biederman et al. [13] also examined kidney biopsies and found *ACTB* and *YWHAZ* to be the most suitable RGs, with less favorable results for *GAPDH* and beta2-microglobulin, acidic ribosomal protein 36B4, and cyclophilin A. While both studies examined a large pool of samples, they were limited by the nature of qPCR compared to RNA-seq, e.g., having to check each RG individually instead of having access to all sequenced transcripts and the lack of available sequencing data from different renal diseases, which were not available at time.

It is interesting to note that non-microdissected datasets appear to yield less differentially expressed genes compared to the microdissected datasets. However, the microdissected dataset also boasted a considerably larger number of patients, on average, in each dataset, compared to the non-microdissected datasets. In non-microdissected data "noise" from larger compartments might mute differential expression of specific RGs in defined compartments. Therefore, the discrepancy between datasets in differentially expressed RGs might be due to the larger number of patients and nature and quality of samples. The data from the Fabry

a

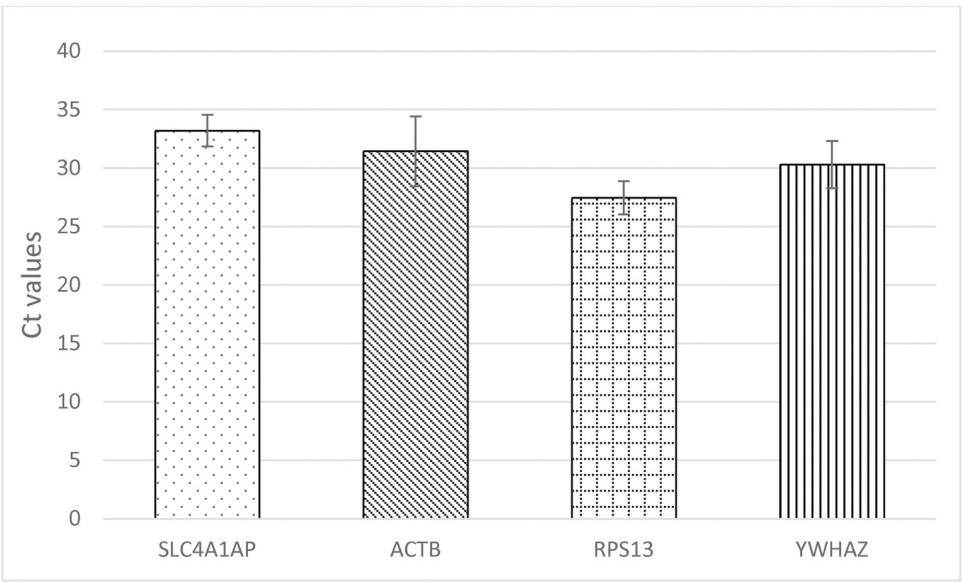

b

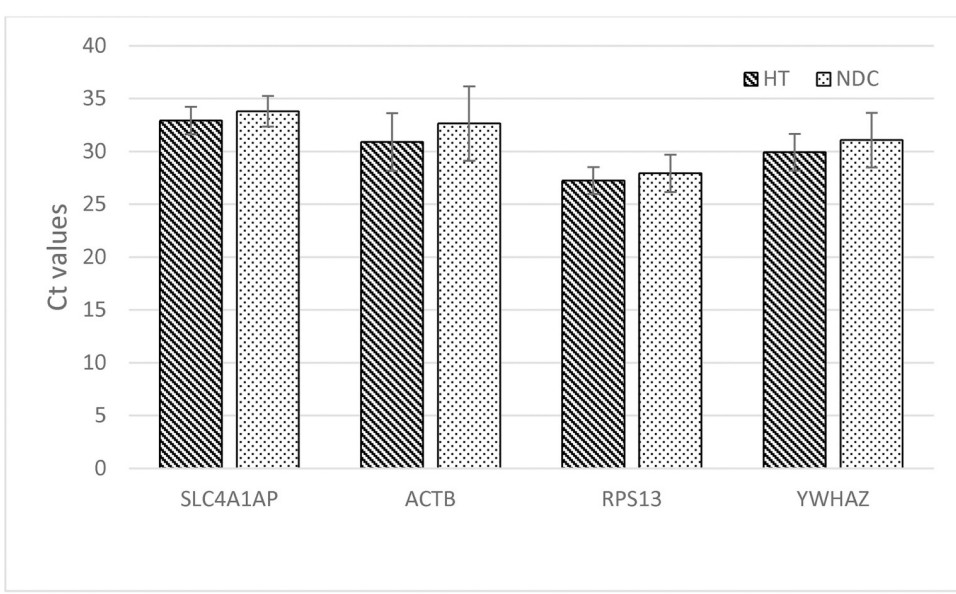

**Fig 4. qPCR Ct values for selected reference genes.** A) qPCR cycles for all 16 samples. B) qPCR cycles for hypertensive (HP; n = 11) and Non-diseased controls (NDC; n = 5). None of the displayed results was significant at the $p < 0{,}05$ level. The y-axis displays Ct values directly. HT = hypertensive group, NDC = non-diseased controls. Data are represented as Mean±SD.

**Table 3. Comparison of the expression of each reference gene in HT and non-diseased control biopsies.** **3a** displays the pvalues from the qPCR experiments. Data were analyzed by Mann-Whitney Asymp. Sig. (2-tailed). None of the comparisons yielded statistically significant results. **3b** shows the fold change (FC) differences and Pearson's correlation in the expression of selected references genes. Fold changes are represented as mean±SD log FC.

Table 3a.

| Reference Gene Candidates | RPS13 | ACTB | SLC4A1AP | YWHAZ |
|---|---|---|---|---|
| **p-value** | 0,336 | 0,282 | 0,336 | 0,336 |

Table 3b.

| | RPS13 | ACTB | SLC4A1AP | YWHAZ |
|---|---|---|---|---|
| **RPS13** | 1 | 3,973±1,708 0,948** | 5,743±0,628 0,899** | 2,827±0,731 0,968** |
| **ACTB** | | 1 | 1,771±1,858 0,933** | 1,146±1,207 0,959** |
| **SLC4A1AP** | | | 1 | 2,916±0,927 0,922** |
| **YWHAZ** | | | | 1 |

**p-val<0,00001.

dataset especially, yielded many differentially expressed RGs. In particular *GAPDH*, SLC4A1AP, *PPIA* and *ACTG1* were only differentially expressed in the Fabry datasets. However, the Fabry datasets were also the only ones including microdissected arteries and differentiating proximal from distal tubules, whereas other datasets referred to either glomeruli or whole tubulointerstitium samples. Thus, the number of differentially expressed RGs might simply reflect true differences that are normally concealed in datasets based on less discriminating whole-section based sequencing.

Methodologies used to study RGs' expression such as microarray, RNAseq or qPCR might produce skewed results, when compared to each other, due to biases intrinsically associated to defined technologies. A contraindicative argument against the mentioned statement could be represented by the largely concurrent expression of defined RG, such as *GAPDH* [7, 11, 13]. However, already in a study from 2003, based on the analysis of 165 microdissected renal biopsies obtained from a variety of diseases, Schmid et al. showed that *GAPDH*, though historically frequently used [7], displays a remarkable variety in its expression level and is thus not suitable as an RG in renal tissues, as also shown in studies on renal cell carcinoma [34, 35].

A similar case of concurrent results between independent sequencing and qPCR data could made for YWHAZ, which proved one of the most suitable RGs investigated in this study and yielded similar results in a separate investigation into microdissected diabetic glomeruli [13].

However, while some results obtained by sequencing and qPCR do concur, others do not. In their study leveraging the massive data contained in The Cancer Genome Atlas (TCGA), Jihoon Jo et al. [11] discarded most of the historically used RGs, such as *GAPDH* or *ACTB* and identified and confirmed by qPCR several new RGs. However, some of their proposed RGs, *HNRNPL*, *PCBP1* and *RER1* appear to be differentially expressed in several of our own datasets. A possible explanation could reside in the focus of this study on cancerous tissue [11]. This again shows that caution should be taken in using RGs validated in one type of tissue, or even just a different disease type, and using them in a different type or disease. Jihoon Jo et al. leveraged an enormous number of samples, but since they were not from non-cancerous renal tissue their results do not apply to that tissue, even though they examined renal biopsies.

Another question regarding RGs and these two techniques is whether an RG suitable for qPCR is also suitable for sequencing via microarray or next-generation sequencing techniques.

An additional level of complexity might not only be related to "true" variability of the levels of defined gene expression, but also to insufficiently specific measurement methods. Veres-Szekely et al. [6] demonstrated that primer specificity is crucial when using *ACTB* as an RG. Unspecific primers might erroneously attach to α-SMA gene, which is upregulated in

Fibroproliferative diseases. As kidney disease and failure are frequently associated with the presence of fibrotic tissue, this might represent an important issue.

Variation of RG expression was previously investigated in a variety of renal cell lines and in renal biopsies from malignant or non-cancerous tissues [7, 13, 34, 35]. However, our study takes advantage of the access to a large number of different datasets, both our own and from independent groups, including samples from over 10 common renal diseases and both micro-dissected and non-microdissected biopsies. Moreover, although not representing an exhaustive list of all RGs that have been, or are, in use, our selection covers a broad range of genes, including older, frequently used RGs, and newer, more recently proposed, candidates. In addition, we further supported our results by performing our own qPCR experiments solely focused on recording RG variability in HT biopsies.

Limitations of our study should also be acknowledged. First, although we comparatively analyzed 30 datasets, 8 were from patients with Fabry's disease. This may have placed an undue influence on the expression of our selected RGs in Fabry disease compared to more common causes of renal failure, such as hypertension. Additionally, several datasets included relatively few patients. Also, we did not distinguish between results garnered from datasets with large, compared to small, populations. Lastly, our cohort for PCR validation was relatively small. However, the acquisition of kidney biopsies, especially from healthy patients, is not as easy as the acquisition from cancerous tissue during, e.g., nephrectomy. Especially as the procedure is not without risk to the patients' health.

## 5. Conclusion

Our analysis suggests that RG suitable to all techniques and tissues do not exist and that they must be carefully selected according to the characteristics of available specimens. Even microdissected tissues might require a separate RG for each compartment, as previously proposed [36]. In non-cancerous kidney biopsies however, we propose that expression of *YWHAZ* as a stable single gene or the combination of *YWHAZ* and SLC4A1AP genes might be of particular interest for normalization purposes, especially in qPCR experiments.

## Supporting information

**S1 Table. Full results with pvalues and foldchanges.** MCD: Minimal change disease, MN: Membranous nephropathy, HT: Hypertension, DIA2: Diabetes type 2, FSGS; Focal segmental glomerulosclerosis, IGAN; IgA nephropathy, TCMR: t-cell mediated rejection, RPGN; Rapidly progressive glomerulonephritis, STA: stable allograft, ATI: acute tubular injury, IFTA: Interstitial fibrosis and tubular atrophy. In the columns under the gene IDs "Yes" refers to genes differentially expressed in control and test samples in the dataset. "No" refers to RG equally expressed in control and test samples in the defined dataset. Not available (NA) refers to RG not tested in specific datasets. Not detected (ND) refers to genes undetected in the specific dataset. Summaries and percentages are noted below each column.
(XLSX)

## Acknowledgments

We are grateful to Celine C. Berthier for her valuable suggestions and assistance in data acquisition for this manuscript. We are also grateful to Giulio Spagnoli for providing language editing services.

## Author Contributions

**Conceptualization:** Philipp Strauss, Håvard Mikkelsen, Jessica Furriol.

**Data curation:** Philipp Strauss, Håvard Mikkelsen, Jessica Furriol.

**Formal analysis:** Philipp Strauss, Håvard Mikkelsen, Jessica Furriol.

**Investigation:** Philipp Strauss, Håvard Mikkelsen.

**Methodology:** Philipp Strauss, Jessica Furriol.

**Project administration:** Philipp Strauss, Jessica Furriol.

**Resources:** Philipp Strauss.

**Supervision:** Philipp Strauss, Jessica Furriol.

**Validation:** Philipp Strauss, Håvard Mikkelsen.

**Visualization:** Philipp Strauss, Håvard Mikkelsen.

**Writing – original draft:** Philipp Strauss, Håvard Mikkelsen.

**Writing – review & editing:** Philipp Strauss, Håvard Mikkelsen, Jessica Furriol.

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
