## [Decision Letter · Decision Letter 0]

16 Jun 2021

PONE-D-21-13929

Variable expression of eighteen common housekeeping genes in human non-cancerous kidney biopsies

PLOS ONE

Dear Dr. Strauss,

Thank you for submitting your manuscript to PLOS ONE. After careful consideration by 2 Reviewers and an Academic Editor, all of the critiques of both Reviewers, especially Reviewer #2, must be addressed in detail in a revision to determine publication status. If you are prepared to undertake the work required, I would be pleased to reconsider my decision, but revision of the original submission without directly addressing the critiques of the 2 Reviewers does not guarantee acceptance for publication in PLOS ONE. If the authors do not feel that the queries can be addressed, please consider submitting to another publication medium. A revised submission will be sent out for re-review. The authors are urged to have the manuscript given a hard copyedit for syntax and grammar.

**Comments to the Author**

1. Is the manuscript technically sound, and do the data support the conclusions?

Reviewer #1: Yes

Reviewer #2: Partly

2. Has the statistical analysis been performed appropriately and rigorously? 

Reviewer #1: Yes

Reviewer #2: No

3. Have the authors made all data underlying the findings in their manuscript fully available?

Reviewer #1: Yes

Reviewer #2: Yes

4. Is the manuscript presented in an intelligible fashion and written in standard English?

Reviewer #1: Yes

Reviewer #2: Yes

5. Review Comments to the Author

Reviewer #1: In the study by Strauss et al, authors were identified suitable reference gene for non-cancerous renal tissue using available datasets and then the results were validated using renal biopsies. They confirmed the previous results that there is no single global reference gene. They also identified YWHAZ and SLC4AIAP as a suitable reference gene.

1- The quality of figures must be improved, I was not able to see the content of figure 2.

2- It seems the line numbers are misplaced on table 3a and it seems those numbers are part of the table.

Line 138. Add reference of the dataset

Line 133-134” included microdissected tissues from glomeruli, arteries, proximal or distal tubules, and tubointerstitial structures” how did authors compared these dataset together as the initial samples are different?

Line 221. “test samples were tallied and RGs with the lowest number picked as top candidates”. What is the lowest number that authors used ?

Line 301-302: “appeared to be peculiarly concerned” why is that? Elaborate more.

Figure 3a: it is not clear which are 10/30 datasets that expression of different tested RG did not show variation.

Figure 3b: those missing bar in the figure is it due to low expression level or the expression is zero? You should modify the Y axis to present this part.

Reviewer #2: The study looks to verify housekeeper genes for relative quantification of mRNA .

This is in itself an important question as having a gene to normalize against that controls for tissue input it self and is not affected by the process been studied and allows for measurement of fold expression of another mRNA species is very important. Many commonly used housekeeping genes have been poorly validated leading to erroneous conclusions regarding expression of mRNA between control and experimental groups.

Validation of housekeeper genes in huma studies is not easy as well defined true normal groups are not easy to access.

The study ask some worthwhile questions but lacks some clarity and should demonstrate the effect of different RG with different genes of interest in various tissues.

RNA seq does not use RGs and while chip analysis may do it needs to be specified when and how in terms of this study. How did the ranking and variability of the refence genes perform across the various techniques.

Some more specific questions

One presumes biopsies are done for a good reason I.e the patient has an illness. So where do the non disease control specimens come from. This needs very clear specification.

It would be helpful if this was more clearly articulated.

There is an issue with taking tissue from formalin fixed tissue for mRNA extraction rather than rapidly placing tissue into a good RNA preservation solution such as RNA or snap freezing. While it I understood there a difficulties in collecting human tissue in this way it is still the best method. One realises a number of studies have done it from formalin fixed tissue to what extent might the various RNA species degrade differently and affect the findings?

The area size of the tissue blocks is mentioned but not region of kidney tissue. This may or may not affect housekeeper genes but would highly likely affect region of nephron structures an vasculature.

406 g is a relatively large amount of RNA from a tissue sample unless relatively large.

Was this total RNA or poly mRNA .

What was the relative purity of the RNA ?

For qPCR how many plates were run and how was plate to plate variability accounted for ?

The Benjamini-Hochberg method is usually used for relatively large data sets such as those used in RNA seq and other large data sets. In analysing the qPCR data for the n=16 patients it is unclear why the various established methods of determining gene variance and hence stability within and between groups was not used. This needs to be explained and justified.

It is now generally accepted that some form of normalization of RNA seq data is required. This would be particularly true for many of the databases examined here. There is a lack of clearly indicated information on stating material. If all that is available is the broad diagnosis then should be indicated. There are varying assumptions that can be made regarding how normalization can be done and violation of the assumptions can lead to erroneous conclusions.

As this study is all about defining genes with relatively constant expression for normalization these assumptions should be spelled out and it made clear how all samples fall within the those assumptions and what might be expected such as very high and low expressing genes. (Evans et al Briefings in Bioinformatics, 19(5), 2018, 776–792) Aanes at al 2014 ) PLoS ONE 9(2): e89158.

In the description of the various RNA seq procedures there is discussion of analysis of differentially expressed gene’s and genes with greater than 2 fold expression excluded yet the paper is about genes that are hopefully minimally differentially expressed. This seems more a generic discussion of RNA seq rather than specific to this study.

As I can see RNA seq is been used as useful way to determine gene expression of a number of genes at once across a variety of samples and should in principle provide a valuable set of relative gene expression data to determine within and between group variation and thus suitability as potential housekeepers. Such an analysis seems lacking.

For the microarray data how was normalization done an what was relatively gene stability and what might have that don in assessing some variable mRNA species of interest.

I think some tables of expression data from the RNA seq and latter qPCR with a clear ranking would be helpful.

I think it would be helpful to determine what might be any experimental bias by clearly showing the effects of using some gene of interest that might change between conditions and how this can be affected by using a single or pair of particular housekeeper genes and how the results might vary between RNA seq and qPCR.

The authors should clearly show the effect of using various combinations of housekeeper genes.

Lines 412 to 414in discussion

While it is true rt-PCR requires each gene to be measured individually why would it require investigators to be more selective in diseases studied. I would suggest that it is far more likely RT PCR would be used than next gene RNA-seq based on cost and availability.

Depending on the nature and extent of disease type of tissue and region of tissue it is hardly surprising there are differences in number of genes expressed. In the kidney it is well described that there is differential gene expression along the nephron and I would expect vasculature. Another variable would be how tissue was dissected and time taken for tissue fixation and differential stability of RNA species.

Whole tissue blocs would be expected to have less variability but again will be subject to sampling and how a piece of tissue is cut. It is also reasonable that the number of patients in each data set might be important. With that in mind a full racial and demographic reporting including drug and co existent morbidities needs to be reported as many conditions might affect renal gene expression.

The authors mention possible differences in methodologies of studying reference or housekeeper genes. RNA-seq by its nature usually does not use RG’s but is dependent on sequencing and counting to normalize samples. Again it would have been useful to see some analysis of variation in the expression of suggested housekeepers by RNA-seq using the various published approaches.

The discussion could be more tightly focused.

This is about RG’s

The abstract mentions looking at RG’s patients with hypertensive nephropathy yet there is little clear data presentation o this subject or what effects different refence genes might have on assessing expression of some other gene of interest.

IT seems that some more comment might have bene made about using different reference genes for different circumstances.

6. PLOS authors have the option to publish the peer review history of their article (what does this mean?). If published, this will include your full peer review and any attached files.

**Do you want your identity to be public for this peer review?** For information about this choice, including consent withdrawal, please see our Privacy Policy.

Reviewer #1: No

Reviewer #2: No

We look forward to receiving your revised manuscript.

Kind regards,

Stephen D. Ginsberg, Ph.D.

Section Editor

PLOS ONE

Journal Requirements:

2. Please provide additional details regarding participant consent. In the ethics statement in the Methods and online submission information, please ensure that you have specified whether consent was informed.

3. In your Methods section, please provide additional information about the participant recruitment method and the demographic details of your participants. Please ensure you have provided sufficient details to replicate the analyses such as: 

a) the recruitment date range (month and year), 

b) a description of any inclusion/exclusion criteria that were applied to participant recruitment, 

c) a table of relevant demographic details, 

d) a statement as to whether your sample can be considered representative of a larger population, and 

e) a description of how participants were recruited.

---

## [Author Response · Author response to Decision Letter 0]

6 Sep 2021

Bergen, September 2021

Editorial Board

PLOSONE 

Dear editor and reviewers

Thank you for taking the time to review our work. We have taken your comments to heart and improved the manuscript accordingly. Below you will find a point-by-point review of the changes to the paper in response to each comment. Each point made by the reviewer is listed first, followed by a reply or explanation and reference to where in the text the changes made to the manuscript are to be found. 

Reviewer #1: 

In the study by Strauss et al, authors were identified suitable reference gene for non-cancerous renal tissue using available datasets and then the results were validated using renal biopsies. They

confirmed the previous results that there is no single global reference gene. They also identified YWHAZ and SLC4AIAP as a suitable reference gene.

1- The quality of figures must be improved, I was not able to see the content of figure 2.

Thanks for the suggestion. We have improved the quality of Figure 2 within the manuscript, see figure 2. 

2- It seems the line numbers are misplaced on table 3a and it seems those numbers are part of the table.

We have fixed the issue.

Line 138. Add reference of the dataset

The references and the GEO numbers of the datasets are listed in table 1, we have amended the text to make this clearer, see line 137 and Table 1. 

Line 133-134" included microdissected tissues from glomeruli, arteries,

proximal or distal tubules, and tubointerstitial structures" how did

authors compared these dataset together as the initial samples are

different?

We thank the reviewer for pointing out this lack of clarity, we have made changes to better explain our methodology. In short, all comparisons were kept within each dataset and within each compartment. So e.g., glomeruli from hypertensive patients from dataset 3 were only ever compared to glomeruli from other groups in dataset 3 e.g., Normotensive controls. We did not compare groups across datasets. See line 155-159.

Line 221. "test samples were tallied and RGs with the lowest number

picked as top candidates". What is the lowest number that authors used ?

The lowest number is zero, i.e. An RG which was not differentially expressed in any of the datasets examined and thus a very stable candidate. We have updated the manuscript for greater clarity, see Line 227 – 228. 

Line 301-302: "appeared to be peculiarly concerned" why is that?

Elaborate more.

Thanks for pointing this out. We obtained three datasets that fulfilled the criteria mentioned in the text (non-microdissected and using stable allografts as controls) and none of the three detected YWHAZ or SLC4AIAP. Though all three datasets originated from the same experiment, so they are not independent from each other. See lines 308-310.

Figure 3a: it is not clear which are 10/30 datasets that expression of different tested RG did not show variation.

Figure 3b: those missing bar in the figure is it due to low expression

level or the expression is zero? You should modify the Y axis to present

this part.

Thanks for your very useful comment. After discussing it, we think that figures 3a and b were not clear enough and we have changed them. We hope the new figures are more representative and easier to understand, see Figure 3 and lines 322 – 332. 

Reviewer #2: 

The study looks to verify housekeeper genes for relative

quantification of mRNA .

This is in itself an important question as having a gene to normalize against that controls for tissue input it self and is not affected by the process been studied and allows for measurement of fold expression of another mRNA species is very important. Many commonly used housekeeping genes have been poorly validated leading to erroneous conclusions regarding expression of mRNA between control and experimental groups.

Validation of housekeeper genes in huma studies is not easy as well defined true normal groups are not easy to access. The study ask some worthwhile questions but lacks some clarity and

should demonstrate the effect of different RG with different genes of interest in various tissues.

We thank the reviewer for his comments. On the issue of using various tissues, this study examines 5 types of kidney specific tissue: whole-section, arteries, glomeruli, distal tubule, and proximal tubule. While additional types of tissue might have been interesting, our area of interest is nephrology and we do not have access to tissue banks from other organs. See line 153 - 156 and Table 1

On the issue of using ‘’different RG with different genes of interest’’: We agree that an analysis of the effect of using various RGs on theoretical genes of interest would be interesting, however, it is outside the aim of this paper. Our aim was to validate proposed RGs using available data by examining their intragroup stability, not a separate analysis of the effects of various normalization strategies on qPCR results. 

RNA seq does not use RGs and while chip analysis may do it needs to be specified when and how in terms of this study. 

We thank the reviewer for this observation. As the reviewer points out RNAseq does not utilize reference genes, but a section in our introduction gave that impression. We have amended that section for greater clarity, see line 55 – 56. 

How did the ranking and variability of the refence genes perform across the various techniques. 

This is an interesting question, we have added some information to the manuscript to better study this angle. We examined the gene expression differences with 3 techniques: RNAseq, microarray (MA) and, for a selection of RGs, qPCR. In qPCR all the 4 candidates were stable, even though the RPS13 gene had been unstable in several RNAseq datasets, so there is some variation across techniques, though cohort size is an important factor. In order to make it simpler to examine variation across techniques we have modified supplementary Table 1. We have included the technique used in each dataset and the amount of variation for each dataset has also added. Figure 2 has also been modified for easier understanding. Overall, the average amount of RGs differentially expressed across all RNAseq datasets was 3.25, while the same number was 1.6 across MA datasets. However, MA datasets were also more often microdissected. See Table S1, Table 1, Figure 2 and line 427 – 445. 

Some more specific questions 

One presumes biopsies are done for a good reason I.e., the patient has an illness. So where do the non-disease control specimens come from. This needs very clear specification. It would be helpful if this was more clearly articulated.

An exceedingly important question. We have modified the paper for additional information. We can’t answer for external datasets, beyond what is available in the respective publications and their GEO pages, though they often use pre-transplantation biopsies. In our own data normal controls were selected from a group of biopsies graded by the renal-pathologist on duty as ‘’not containing any or only insignificant pathology’’. We then re-examined the biopsy, verified the non-pathological histology and accessed the patients clinical record for further information. Patients that later developed renal disease, severe autoimmune disease or showed severe proteinuria were excluded. The exact clinical reasoning for taking the biopsy in the first place, and underlying pathology, if any was found, varied substantially from patient to patient. See line 140 – 150.

There is an issue with taking tissue from formalin fixed tissue for mRNA extraction rather than rapidly placing tissue into a good RNA preservation solution such as RNA or snap freezing. While it I understood there a difficulties in collecting human tissue in this way it is still the best method. One realises a number of studies have done it from formalin fixed tissue to what extent might the various RNA species degrade differently and affect the findings?

Results for the same gene seem to vary very little between fresh-frozen and FFPE storage, Eikrem et.al. compared FFPE to fresh frozen samples from the same biopsy and the average expression and the log2 fold changes of these transcripts correlated with R2 = 0.97, and R2 = 0.96, respectively. (PLoS One. 2016 Feb 22;11(2):e0149743. doi: 10.1371/journal.pone.0149743. eCollection 2016.’’Transcriptome Sequencing (RNAseq) Enables Utilization of Formalin-Fixed, Paraffin-Embedded Biopsies with Clear Cell Renal Cell Carcinoma for Exploration of Disease Biology and Biomarker Development’’). 

Similar results have been found in several other studies; see for example Esteve-Codina A, Arpi O, Martinez-García M, Pineda E, Mallo M, Gut M, et al. (2017) A Comparison of RNA-Seq Results from Paired Formalin-Fixed Paraffin-Embedded and Fresh-Frozen Glioblastoma Tissue Samples. PLoS ONE 12(1): e0170632. 

Or Bossel Ben-Moshe, N., Gilad, S., Perry, G. et al. mRNA-seq whole transcriptome profiling of fresh frozen versus archived fixed tissues. BMC Genomics 19, 419 (2018). https://doi.org/10.1186/s12864-018-4761-3

Though it should be mentioned that not all species of RNA are as easily found in FFPE as in FF tissue, such as soluble RNA etc. However, RNAs that is present in FFPE tissue seem to behave similar to fresh frozen tissue. 

The area size of the tissue blocks is mentioned but not region of kidney

tissue. This may or may not affect housekeeper genes but would highly

likely affect region of nephron structures an vasculature.

Another good question. We have amended the manuscript. The answer to this comment overlaps a bit with another comment on page 11 (‘’ Depending on the nature and extent …’’). We cannot provide additional details on the histological consistency of biopsies in external dataset beyond their peer reviewed status and the information provided in the original publication. In our own data biopsies were always taken for diagnostic purposes and therefore aimed at the kidney cortex. Biopsies with less than 50% cortex were discarded. Approximately 70% of the biopsies had 10 or more glomeruli. All biopsies contained arterioles to some degree while larger vessels were very rare. 

Microdissected samples of course only contain their respective compartments. 

See lines 144-147.

406 g is a relatively large amount of RNA from a tissue sample unless relatively large.

The yield was given as nanogram, not gram. The section has been amended for greater clarity, see lines 199 – 200.

Was this total RNA or poly mRNA.

Total RNA, see lines 201 – 202. 

What was the relative purity of the RNA?

Thank you for the comment. 260/280 ratio had a median of 1,905 (range 1,67-1,98) and ratio 260/230 of 1,85 (1,01-2,11). We have added the data within the manuscript. See lines 200 – 204. 

For qPCR how many plates were run and how was plate to plate variability

accounted for?

We run 3/4 technical replicates for each sample and each probe. We have added it to the manuscript. We agree that accounting for plate-to-plate variability is very interesting. Unfortunately, we had not enough material to replicate the analysis, and, as we were mainly interested in the differences between hypertensive nephropathy and controls, we decided to only perform technical replicates. See lines 209 – 210. 

The Benjamini-Hochberg method is usually used for relatively large data sets such as those used in RNA seq and other large data sets. In analysing the qPCR data for the n=16 patients it is unclear why the various established methods of determining gene variance and hence

stability within and between groups was not used. This needs to be explained and justified.

Thank you for the suggestion. We have improved the explanation in the manuscript. We only used Benjamini- Hochberg method in the 30 RNA datasets that we used to assess the differential gene expression of the genes of interest. For qPCR, we analyzed the data using the Mann-Whitney U test. See line 231 – 232.

It is now generally accepted that some form of normalization of RNA seq data is required. This would be particularly true for many of the databases examined here. There is a lack of clearly indicated information on stating material. If all that is available is the broad diagnosis then should be indicated. There are varying assumptions that can be made regarding how normalization can be done and violation of the assumptions can lead to erroneous conclusions.As this study is all about defining genes with relatively constant expression for normalization these assumptions should be spelled out and it made clear how all samples fall within the those assumptions and what might be expected such as very high and low expressing genes. (Evans et

al Briefings in Bioinformatics, 19(5), 2018, 776-792) Aanes at al 2014 ) PLoS ONE 9(2): e89158.

Great Comment.

Regarding starting material, this is explained in greater detail in a different comment on page 12 (‘’ Whole tissue blocks…’’) and a comment on page 6 (‘’ The area size of the tissue blocks…’’). We have answered the concerns regarding starting material below those comments. 

Regarding normalization, we have used CPM-normalization and a DESeq2 normalization for our internal datasets. For the external datasets, we have used the original authors own normalization, with the information available in the original publication and GEO submission. When no normalized data was available, we have used the same normalization strategies as for our own data. No data was compared between groups; all comparisons were kept within each dataset and within each compartment. So e.g., glomeruli from hypertensive patients from dataset 3 were only ever compared to glomeruli from other groups in dataset 3 e.g., Normotensive controls. We did not compare groups across datasets. So, while normalization between datasets may vary, it does not impact the results. The biological scaling normalization (BSN) referred to in the paper the reviewer linked was tested in embryonic zebrafish, where the amount of RNA was significantly different based on the embryonic development (‘’ The first 2.5 h are characterized by substantial increase of polyA+ RNA, while there is massive decay of RNA due to miRNA-430 activation at 3.5 hpf and onwards’’). However, in our case we are comparing biopsies from healthy human adults, so substantial differences due to ongoing development are not present. Additionally, the authors themselves point out that ‘’this illustrates the key difference between the normalization methods compared; BSN seek to maintain biological differences, while RPM and TMM lead to samples with similar distribution of the gene expression levels.’’ As such RGs that were differentially expressed using the standard normalization methods would most likely also be differentially expressed using the BSN normalization. 

See lines 155-159.

In the description of the various RNA seq procedures there is discussion of analysis of differentially expressed gene's and genes with greater than 2 fold expression excluded yet the paper is about genes that are hopefully minimally differentially expressed. This seems more a generic

discussion of RNA seq rather than specific to this study. 

We agree that other, lower, thresholds might have been chosen, which might have yielded a greater number of RGs with differential expression. However, we believe that a threshold of 2-fold expression is suitable as some, minimal, variation has to be condoned even in RGs, due to the lack a gene with a complete lack of variation and expected standard error of the methodology. 

As I can see RNA seq is been used as useful way to determine gene expression of a number of genes at once across a variety of samples and should in principle provide a valuable set of relative gene expression data to determine within and between group variation and thus suitability as potential housekeepers. Such an analysis seems lacking. 

We thank the reviewer for this comment and have made changes to the paper to answer them. Also please note that the answer to this comment overlaps to some extent with the answer to a previous comment on page 4 (‘’ How did the ranking and variability of the refence genes…’’)

In short; Table S1 displays the direction of changes for each RG for each dataset, and can thus provide information on inter-experiment variation, e.g. the direction of the foldchange for a specific RG across experiments, groups and methodology. See Table S1.

For the microarray data how was normalization done an what was relatively gene stability and what might have that don in assessing some variable mRNA species of interest. 

All microarray data were obtained externally, from different groups. As such the exact method for normalization differed, depending on the original submitter. However, this information is available for each dataset through the GEO submission, which is referenced in each case. Dataset 9 and 10, for instance, were normalized as follows, according to information provided through the original GEO submission (GEO GSE104948). 

‘’…Arrays were RMA normalized using probe sets common to U133A and U133 Plus2.0 and batch corrected using COMBAT (ERCB)…’’ 

We felt it unnecessary to reiterate the normalization process in cases where this information was already stated elsewhere, especially if it was part of an original submission. 

We used adj. pvalues as proxies of gene stability, the same as for our own data. A gene with an adj. pvalue below 0.05 (for differential expression) was considered as unstable and thus unsuitable as an RG. See lines 235-236 and 287 – 292. 

I think some tables of expression data from the RNA seq and latter qPCR with a clear ranking would be helpful.

The supplementary table S1 has foldchanges for every reference gene in every dataset. Additionally, we also display the adjusted pvalues for every reference gene in every dataset. Similar data for the qPCR results are displayed in Table 3.

The pure read counts are also available, for the external datasets as part of the GEO submission in question and our own data has been uploaded to GitHub, repository 310590-transciptomic-data. See lines 519 - 521, Table 3 and Table S1. 

I think it would be helpful to determine what might be any experimental bias by clearly showing the effects of using some gene of interest that might change between conditions and how this can be affected by using a single or pair of particular housekeeper genes and how the results might

vary between RNA seq and qPCR. The authors should clearly show the effect of using various combinations of housekeeper genes.

Thank you for your suggestion. In the article, we do not try to suggest different sets of housekeeping genes, but rather verify established housekeeping genes based on our own and publicly available datasets. To mitigate condition bias, we performed qPCR on some of the housekeeping genes to confirm our findings from the different expression datasets.

We agree that an analysis of the effect of using various RGs on a number of theoretical genes of interest would be interesting. However, it is outside the scope of this paper. Our aim was only ever to validate proposed RGs using available data by examining their intragroup stability, not a separate analysis of the effects of various normalization strategies on qPCR results. 

Lines 412 to 414in discussion

While it is true rt-PCR requires each gene to be measured individually

why would it require investigators to be more selective in diseases

studied. I would suggest that it is far more likely RT PCR would be used

than next gene RNA-seq based on cost and availability.

We agree that this point was poorly explained, and we thank the reviewer for pointing it out. We have amended the section. Our original point was this: for checking individual, or a low number of genes, qpcr is superior. The advantage of RNAseq here lies in sequencing every gene simultaneously, without focusing on any one gene in particular. Since data from additional renal diseases is often available online, researchers don’t have to constrain themselves to a few genes or a few diseases. See lines 424-425. 

Depending on the nature and extent of disease type of tissue and region

of tissue it is hardly surprising there are differences in number of

genes expressed. In the kidney it is well described that there is

differential gene expression along the nephron and I would expect

vasculature. Another variable would be how tissue was dissected and time

taken for tissue fixation and differential stability of RNA species.

We thank the reviewer for this insightful comment.

The answer to this comment overlaps to some extent with a previous comment on page 6 (‘’ The area size of the tissue…’’). Regarding tissue region, we cannot provide additional details on the histological consistency of biopsies in external dataset beyond their peer reviewed status and information provided as part of the original population. In our own data, biopsies were initially taken for diagnostic purposes and therefore aimed at the kidney cortex. Biopsies with less than 50% cortex were discarded. Approximately 70% of the biopsies had 10 or more glomeruli. All biopsies contained arterioles to some degree while larger vessels were very rare. As such all biopsies were vascular to some degree, with minor variations. 

Microdissected samples of course only contain their respective compartments, and microdissection was performed by personnel with long experience in renal histology. In cases where we were unsure of whether a structure was e.g. A proximal or distal tubule that section of the slide was excluded. 

See lines 144-147.

Concerning sample processing; Again, we cannot answer for procedures performed during processing of external data, beyond their peer-reviewed status and the details that the original author has provided. For biopsies taken for our own data after obtaining the biopsy the tissue was immediately handed to a technician who fixated the biopsy as formalin-fixed paraffin embedded tissue, without delay. 

All microdissections were performed on the same Zeiss PALM Lasor Capture Microdissection (LCM) system (Carl Zeiss AG, Oberkochen, Germany) with consistent personal and settings for each dataset. After microdissection the samples were immediately stored at -80 degrees till rna extraction, after which they were again immediately stored at -80 degrees. 

See lines 147 – 150. 

Whole tissue blocs would be expected to have less variability but again

will be subject to sampling and how a piece of tissue is cut. It is also

reasonable that the number of patients in each data set might be

important. With that in mind a full racial and demographic reporting

including drug and co existent morbidities needs to be reported as many

conditions might affect renal gene expression.

Also a good suggestion. We have greatly expanded the amount of information provided on the cohorts. See line 164 – 170. 

The authors mention possible differences in methodologies of studying

reference or housekeeper genes. RNA-seq by its nature usually does not

use RG's but is dependent on sequencing and counting to normalize

samples. Again it would have been useful to see some analysis of

variation in the expression of suggested housekeepers by RNA-seq using

the various published approaches.

We thank the reviewer for this comment and have made changes to the paper to answer them. As the reviewer points out, this point has been brought up before, in a comment on page 4 (‘’ How did the ranking and variability of the refence genes…’’). We have answered in full there. 

In short; Information on the variation in the expression of suggested housekeepers is displayed in Table S1; the foldchanges for every RG in every dataset (540 datapoints), analysis of variation for each dataset and RG as a whole are also displayed. See Table S1

The discussion could be more tightly focused. This is about RG's

The abstract mentions looking at RG's patients with hypertensive

nephropathy yet there is little clear data presentation on this subject

or what effects different refence genes might have on assessing

expression of some other gene of interest.

We have examined non-cancerous kidney biopsies, which includes biopsies from patients with hypertensive nephropathy. We have also looked at biopsies from patients with diabetic nephropathy, Fabry disease, focal segmental glomerulosclerosis, IgA nephropathy, membranous nephropathy, and minimal change disease, all of which is stated in the abstract. The patients for the qPCR experiment were also taken from patients with hypertensive nephropathy. See lines 30 – 38 and 181 – 190.

IT seems that some more comment might have bene made about using

different reference genes for different circumstances.

We have added some parts to the discussion regarding the use of different

reference genes for different circumstances. See lines 455- 464.

To conclude this letter, we would like to thank again all reviewers for their valuable comments. We hope to have answered all issues satisfactory.

Sincerely,

Philipp Strauss

University of Bergen

Department of Clinical Medicine

Haukeland University Hospital

Jonas Lies Vei 65,

Laboratory Building, 7th floor

5021 Bergen, Norway

Mobile phone: 0047 93686433

E-mail: Philipp.Strauss@uib.no

---

## [Decision Letter · Decision Letter 1]

27 Sep 2021

PONE-D-21-13929R1Variable expression of eighteen common housekeeping genes in human non-cancerous kidney biopsiesPLOS ONE

Dear Dr. Strauss,

Thank you for resubmitting your work to PLOS ONE. Please make the corrections posed by Reviewer #2 so I can render a decision on this manuscript.

**Comments to the Author**

1. If the authors have adequately addressed your comments raised in a previous round of review and you feel that this manuscript is now acceptable for publication, you may indicate that here to bypass the “Comments to the Author” section, enter your conflict of interest statement in the “Confidential to Editor” section, and submit your "Accept" recommendation.

Reviewer #2: (No Response)

2. Is the manuscript technically sound, and do the data support the conclusions?

Reviewer #2: Yes

3. Has the statistical analysis been performed appropriately and rigorously? 

Reviewer #2: Yes

4. Have the authors made all data underlying the findings in their manuscript fully available?

Reviewer #2: Yes

5. Is the manuscript presented in an intelligible fashion and written in standard English?

Reviewer #2: Yes

6. Review Comments to the Author

Reviewer #2: I think the paper would benefit from an explicit example of what the effect would be on variation in relative gene expression using a a less stable vs more stable reference gene.

This could even be a theoretical discussion just looking at effect of variation in the denominator ie reference gene.

I think this would make the paper eve more impactful and drive the points home about housekeeper gene stability

7. PLOS authors have the option to publish the peer review history of their article (what does this mean?). If published, this will include your full peer review and any attached files.

**Do you want your identity to be public for this peer review?** For information about this choice, including consent withdrawal, please see our Privacy Policy.

Reviewer #2: No

We look forward to receiving your revised manuscript.

Kind regards,

Stephen D. Ginsberg, Ph.D.

Section Editor

PLOS ONE

---

## [Author Response · Author response to Decision Letter 1]

5 Oct 2021

Bergen, October 2021

Stephen D. Ginsberg, Ph.D.

Section Editor

PLOS ONE

Dear editor and reviewers

Thank you for taking the time to review our work a second time. We have taken your comments to heart and improved the manuscript accordingly. Below you will find a point-by-point review of the changes to the paper in response to each comment. 

Reviewer #2: 

1. Reviewer #2: I think the paper would benefit from an explicit example of

what the effect would be on variation in relative gene expression using a a

less stable vs more stable reference gene.

This could even be a theoretical discussion just looking at effect of

variation in the denominator ie reference gene.

I think this would make the paper eve more impactful and drive the points

home about housekeeper gene stability

Response to 1. Reviewer #2: We thank the reviewer for taking the time to review the manuscript again. We have added a section on the suggested topic to the discussion. See lines 401 – 411. 

To conclude this letter, we would like to thank again all reviewers for their valuable comments and taking the time to review the manuscript again. We hope to have answered all issues satisfactory.

Sincerely,

Philipp Strauss

University of Bergen

Department of Clinical Medicine

Haukeland University Hospital

Jonas Lies Vei 65,

Laboratory Building, 7th floor

5021 Bergen, Norway

Mobile phone: 0047 93686433

E-mail: Philipp.Strauss@uib.no

---

## [Decision Letter · Decision Letter 2]

19 Oct 2021

Variable expression of eighteen common housekeeping genes in human non-cancerous kidney biopsies

PONE-D-21-13929R2

Dear Dr. Strauss,

We’re pleased to inform you that your manuscript has been judged scientifically suitable for publication and will be formally accepted for publication once it meets all outstanding technical requirements.

Kind regards,

Stephen D. Ginsberg, Ph.D.

Section Editor

PLOS ONE

Additional Editor Comments: Please address the minor errors pointed out by the Reviewer in the final submission.

**Comments to the Author**

1. If the authors have adequately addressed your comments raised in a previous round of review and you feel that this manuscript is now acceptable for publication, you may indicate that here to bypass the “Comments to the Author” section, enter your conflict of interest statement in the “Confidential to Editor” section, and submit your "Accept" recommendation.

Reviewer #1: All comments have been addressed

2. Is the manuscript technically sound, and do the data support the conclusions?

Reviewer #1: Yes

3. Has the statistical analysis been performed appropriately and rigorously? 

Reviewer #1: Yes

4. Have the authors made all data underlying the findings in their manuscript fully available?

Reviewer #1: Yes

5. Is the manuscript presented in an intelligible fashion and written in standard English?

Reviewer #1: Yes

6. Review Comments to the Author

Reviewer #1: Strauss et al. identified the suitable RG from noncancerous renal tissues using the available data set and their specimen in this revised manuscript. They addressed the importance of the identification of relevant RG based on the methodology and tissue of interest. They identified YWHAZ as compared to ACTB, which historically was used as an RG. The authors have responded well to the comments. Although a few minor issues remain:

Abstract line 27: are widely use, “d” is missing

Line 36: 3 genes are listed and RPS13 is missing

Line 167, other renal, a word is missing. Is it other renal diseases? Or condition?

Line 339, I think it should be 20 datasets as in line 336 is stated 10/30 datasets, so remaining should be 20.

7. PLOS authors have the option to publish the peer review history of their article (what does this mean?). If published, this will include your full peer review and any attached files.

Reviewer #1: No

---

## [Editor Report · Acceptance letter]

1 Dec 2021

PONE-D-21-13929R2 

Variable expression of eighteen common housekeeping genes in human non-cancerous kidney biopsies 

Dear Dr. Strauss:

I'm pleased to inform you that your manuscript has been deemed suitable for publication in PLOS ONE. Congratulations! Your manuscript is now with our production department. 

Kind regards, 

on behalf of

Dr. Stephen D. Ginsberg 

Section Editor

PLOS ONE